# More Than Two Decades of Research on Selective Traditions in Environmental and Sustainability Education—Seven Functions of the Concept

**Per J. Sund [1,2,\*] and Niklas Gericke [2]**

[1] Department of Mathematics and Science Education, Stockholm University, 10691 Stockholm, Sweden
[2] Department of Environmental and Life Sciences, Karlstad University, 65188 Karlstad, Sweden; niklas.gericke@kau.se
[\*] Correspondence: per.sund@mnd.su.se

**Abstract:** This study investigates functions of the concept of selective traditions by means of a qualitative systematic review synthesis of earlier research. The study is based on a review method for integrating qualitative studies and looks for "themes" in or across them. In this case, it is about how the identified publications (twenty-four in total) use the concept of selective traditions. All but two studies stem from the Swedish context. The selective traditions relate to teachers' approaches to the content, methods and purposes of environmental and sustainability education (ESE). Teachers mainly work within one specific selective tradition. Seven different functions were found in the publications of which five are claimed to be valuable for the development of ESE teaching, while the other two functions are useful in monitoring changes and development in ESE teaching. The results are discussed in terms of the consequences for research, practice and teacher education aiming at offering suggestions on how to develop future (transformative) ESE teaching.

**Keywords:** selective traditions; teaching traditions; teaching habits; environmental and sustainability education; functions of teaching; functions of education; ESD teaching approaches

## 1. Introduction

This study investigates functions of the concept of selective traditions by means of a qualitative systematic review synthesis of earlier research. The study is based on a review method for integrating studies and looks for "themes" in or across them [1]. In this case, it is about how studies use the concept of selective traditions as described by different functions. Selective traditions relate to teachers' approaches to the content, methods and purposes of environmental and sustainability education (ESE). Three teaching traditions of ESE have been identified in previous research: the fact-based tradition of conveying facts, the normative tradition that argues for certain values and lifestyles and the pluralist tradition that focuses on students' participation and emancipation. Teachers mainly work within one specific tradition. However, the traditions are not usually recognized by the teachers themselves, but by researchers using analytical tools. In this study, we identified the specific functions these three selective traditions had been reported to have in previous studies both from an educational and research perspective. The results are discussed in terms of the consequences for research and practice aiming at a systematic development of informed future ESE teaching.

## 2. Background

Research on the teachers teaching different school subjects has shown that they all have different ways, or traditions, of selecting educational content and methods. These traditions can thus be termed selective traditions [2]. Selective traditions can be understood as what teachers consider good teaching. The concept of selective traditions is useful when

discussing environmental and sustainability education (ESE) with teachers in that it is a way of expressing their ambitions to change and develop teaching in a reflected and informed way [3]. The selective teaching traditions of ESE are useful for understanding the role of teachers, students and the purpose of education [4] because it focuses on teachers' responses to the question that is often posed by students, "Why should I learn this?" [5].

The implications for teacher education are also strong [6]. Often, teachers teach in the way they were taught at university; this needs to be recognized and addressed in teacher education so that student teachers are aware of how to change from disciplinary teaching of adults to teaching children and adolescents school subjects [7]. Student teachers need support from teacher educators to change from the focus on disciplinary facts and concepts to also emphasize the importance of students' interests and participation. There is a shift from disciplinary knowledge to everyday knowledge that teacher education needs to pay attention to, and in this, the concept of selective traditions is a useful tool [8].

In the Swedish context, selective traditions have been investigated for more than two decades in science education and environmental and sustainability education research. The most important finding has been the identification of three teaching traditions within environmental and sustainability education: the fact-based tradition, the normative tradition and the pluralistic tradition. These traditions have provided research and practice with an analytical tool that can be used to discuss the role and purpose of education [9] and the students' democratic participation in it [10]. In the following section, we discern these traditions in more detail.

### 2.1. Three Selective Traditions in ESE

Three selective traditions have been identified in environmental education (EE) in Sweden since the 1960s, with reference to educational philosophy and how environmental and developmental problems are understood by teachers [9]. Sandell, Öhman and Östman [9] described three educational philosophies connected to selective teaching traditions: essentialism, progressivism and reconstructivism. The starting points in these three educational philosophies also indicate three different solutions to environmental problems: to the lack of relevant scientific knowledge (facts), to weakly developed attitudes and un-reflected lifestyles (unclear norms) or in the form of informed attempts to solve conflicting human interests (pluralism of solutions).

Selective traditions were studied for the first time in a large study of teachers (*n* = 568) in the Swedish school system by the Swedish National Agency for Education (2002). Teachers mainly work within one tradition. It is important to point out that the descriptions of these traditions (outlined below) were summarized in order to make them easier for the reader to grasp. The traditions are teachers' teaching types. The descriptions outlined below closely follow the original descriptions [3].

The fact-based tradition was formed in the early development of EE. Environmental issues are regarded mainly as ecological issues. Environmental problems are based on the lack of knowledge and can often be solved by science. There is an assumption that if teachers teach scientific knowledge at school, environmental problems will disappear more or less automatically. From the environmental ethics perspective, this tradition lies within modern anthropocentrism. The natural world is considered to be separate from humanity. In terms of educational philosophy, this tradition is closest to essentialism. Essentialism means that the content of education ought to be based on science, that the actual subject matter has priority and that the teaching uses adapted scientific terminology and models. The pedagogic task is to teach pupils the right knowledge and proper knowledge. The teaching style in this tradition is mainly through lectures, with very little group discussion or activities in which the learned knowledge can be applied. Teachers make the planning [9].

The normative tradition emerged during the societal debate in the 1980s, e.g., as a result of the nuclear power referendum in Sweden. Environmental issues are primarily a question of values, where people's lifestyles and their consequences become the main

threats to the natural world. Scientific knowledge can offer hints about the good ways of living and be prescriptive in decision-making. According to the teachers of this tradition, right knowledge is assumed to automatically lead to better values that make people want to change their lifestyle. From an ethical point of view, humans are regarded as an indispensable part of nature and should therefore adapt to its conditions; it is a biocentric view. The teaching content is partly organized in a thematic way and requires content from many disciplines. Attention is paid to the use of pupils' everyday experiences and attitudes when creating teaching examples and tasks [9]. The starting point in progressivism puts pupils in the central position, where the teaching is organized in accordance with the needs and interests of the group of pupils.

The pluralistic tradition developed during discussions in the 1990s. An increasing uncertainty about environmental issues and the number of different standpoints in environmental debates (e.g., Rio Summit 1992) are important points of departure for this tradition. Environmental issues are viewed as political problems and are regarded as conflicts between different human interests [10]. Science does not offer guidance on how to act when it comes to solving environmental issues. In this tradition, EE includes the entire spectrum of social and economic development and is replaced with the concept of ESD [11]. The conflict-based perspective of ESD highlights that everyone's view on environmental issues is regarded as being equally relevant. Pluralism is an important starting point for the conduct of teaching in ESD. Pupils develop their abilities to engage in the development of a sustainable society. This suggests that the lessons are reconstructivist in character. Recontructivism emphasizes the role of the school in the democratic development of a future sustainable society. Teaching methods and approaches vary from an individual search for more scientific facts to writing articles or formulating arguments that can be used and published in newspapers.

Other ways of describing selective traditions in other countries can be found as well. Sauvé [12] and Stables [13] described selective traditions in EE in the context of Canada and the UK. Sauvé's starting point is in the contemporary development of a societal environmental consciousness and discourse, while Stables starts by discussing the importance of enhancing nature relations. Vare and Scott [14] described two types of ESD in the UK that have some similarities with selective traditions: ESD 1 and ESD 2. ESD 1 facilitates a change in our ability to deal with the problems of the present and how we live now by promoting behavioral change, a shift in habits or a change in how things are thought about, where the need for this has been clearly identified and socially agreed on. ESD 2 facilitates a change in our ability to deal with an uncertain and unknown future by enabling pupils to think critically about (and beyond) what is known now and what experts say and to test sustainable development ideas [14].

### 2.2. The Importance of Functions in the Research on Selective Traditions in ESE

This is a review study on the use of the concept of selective traditions in ESE research. The qualitative differences in the use of the concept in different publications can be regarded as different themes, which in this article are called functions. These functions are developed across individual studies described in ESE research publications where the concept is used in a similar way.

This study was inspired by Biesta [15], who describes the purpose of education in terms of functions. A function is described by Biesta as an overarching purpose of education that reflects its aim. In the work by Biesta, he identifies three functions of education. The first function is that education has a role to play in pupils' socialization into the society by conveying social, political and cultural values and behavior that aim to preserve a specific democratic society. The second function is that education contributes to pupils' qualifications, thereby advancing their knowledge, skills and competences for their lives in various areas, such as the labor market (different professions), further studies and as citizens. The third function is that education has a role to play in pupils' subjectification.

This is about the emancipation of pupils as humans and providing them with agency as citizens.

Biesta's [15] approach of using different functions to describe the purpose of education inspired this study to discern the functions the research of a specific concept, in this case, ESE selective traditions, has identified. These identified functions can be used as analytical tools, which can inform the analysis and development of future ESE teaching. This means that different functions of the concept of selective traditions can be used to understand how new and future ESE teaching can be better reflected upon and developed in research, practice and teacher education [16].

## 3. Purpose

The overall purpose of this study was to offer researchers and educators a qualitative systematic review of more than two decades of empirical research on selective traditions in environmental and sustainability education research. Here, the different ways of using the concept in research are referred to as functions. The purpose of the study was to investigate how the concept of selective traditions in ESE research had been assigned qualitatively different functions in earlier research. The study's research question is, "Which functions of the concept of selective traditions are discernible in earlier ESE research?"

## 4. Method and Review Design

For the study, a systematic review was used as method. Systematic reviews seek to draw together all known knowledge on a topic area. In this endeavor, study designs incorporating quantitative, qualitative and mixed method studies can be used [1]. In this study, we used qualitative analysis, but the included studies represent both quantitative and qualitative studies. In the analysis of the selected studies, we used thematic analysis looking for "themes" or "constructs" in and across the individual studies and determined their functions [1].

### 4.1. Literature Search

The review began with a systematic search of selected terms and term combinations in databases (ERIC, EBSCO) and Google Scholar [17]. The terms used in the search represented different combinations of the key terms: "habitual teaching" and "EE/ESD/ESE," "selective" and "EE/ESD/ESE," "selective traditions" and "EE/ESD/ESE," "teaching traditions" and "EE/ESD/ESE." All the studies identified from the search were included in the following analysis. The identified publications (twenty-two in total) were journal articles (fifteen), one doctoral thesis, two books, three book chapters and one national report. Twenty studies were conducted in Sweden, one—in the USA/Spain, one—in the Netherlands. Two manuscripts, one book chapter in progress and one article manuscript in review written by the authors of this literature review, were included. These twenty-four publications in total consisted of five theoretical papers and eighteen empirical studies using surveys, interviews (teacher/pupil), focus groups (teachers) and textbooks as primary data from secondary and upper secondary school. The twenty-four publications are listed in alphabetical order below:

- Borg, Gericke, Höglund and Bergman, 2012;
- Borg, Gericke, Höglund and Bergman, 2014;
- Callahan and Dopico, 2016;
- Education, 2002 (national report);
- Gyllenpalm, Wickman and Holmgren, 2010;
- Lidar, Karlberg, Almqvist, Östman and Lundqvist, 2018 (book chapter);
- Lundegård and Wickman, 2007;
- Lundqvist and Sund, 2018;
- Rudsberg and Öhman, 2010;
- Sund, 2008 (book chapter);
- Sund, 2016;

- Sund, in progress (book chapter);
- Sund and Gericke, 2020;
- Sund and Gericke, in review;
- Sund, Gericke and Bladh, 2020;
- Sund and Wickman, 2008;
- Sund and Wickman, 2011a;
- Sund and Wickman 2011b;
- Sandell, Öhman and Östman, 2005 (book);
- Van Driel, Bulte and Verloop, 2008;
- Van Poeck, Östman and Öhman, 2019 (book);
- Öhman, 2004 (book chapter);
- Öhman and Östman, 2019 (book chapter);
- Östman, 1995 (thesis);

*4.2. Analysis of the Publications*

As already indicated, the aim of a systematic review is to look for "themes" in and across individual studies to extrapolate new general meaning from the included studies [1]. The analytical question used to discern the crosscutting themes was, "How is the concept of selective traditions used in the actual publication?" The twenty-four identified publications were read several times and the focus of the analysis was to find crosscutting themes of what function the concept of selective traditions was given.

First, relevant information was extracted from each publication using a coding sheet. Coded information included both descriptive study characteristics and study findings as guided by the review question related to the function of the concept of selective traditions in the study. Tentative themes were identified to obtain the first, preliminary arrangement of the studies and their findings and to prepare for synthesis. Regardless of whether the information was quantitative or qualitative, all coding had to focus on the key concepts as well as concise summaries of the study findings [1]. In some publications, the concept of selective traditions was used in two different ways, but then the analysis focused on describing its main function. This was the way of making the results of the functions more succinct and useful for ESE researchers and approach developers.

Second, the data analysis stage of the synthesis work was done iteratively, by repeatedly and in a cyclical process considering tentative review findings in relation to individual study findings. The publications with similar answers to the analytical question together formed a specific function. Synthesis meetings were alternated with re-readings of the studies. The purpose of the meetings was to test and, if necessary, revise tentative review findings by creating additional abstractions or reformulations.

## 5. Results

Seven different functions were found in the publications analyzed in this ESE research review, of which five are regarded as valuable for the development of ESE research and practice. Two of the functions are interesting for research on changes in teaching emphasis and the distribution of teaching approaches. The functions are presented below but are not listed in any particular order.

*5.1. Combining Educational Philosophy and Environmental Problems in Teaching*

The first function of the concept of selective traditions in ESE is to combine starting points in educational philosophy with the characteristics of environmental and developmental problems. This function offers researchers and practitioners the possibility to reflect on the origins of educational philosophy and the purpose of ESE teaching, namely what is to be learned, how it should be learned and the nature of the sustainability challenges to be addressed. These are fundamental issues to consider when designing ESE teaching approaches.

Some of the reviewed publications [3,9] elaborated on how selective traditions evolved in environmental education in Sweden with reference to their roots in educational philosophy and how environmental and developmental problems are perceived by teachers. The analytical combination of roots in educational philosophy and how teachers perceive environmental problems resulted in the identification of the fact-based, normative and pluralistic selective traditions in EE [3,9]. The concept of selective traditions is a way of understanding how different ways of ESE teaching emphasize student participation, development of students' democratic (communicating, listening, arguing, debating) and critical abilities (analysis, critical approach, pluralism of alternatives) [11]. Similar selective traditions were described for science teaching [18], where socio-scientific issues (e.g., climate change, sustainability, water and food scarcity) were included [19].

### 5.2. Analysing ESE Teaching Empirically

The second function of the concept of selective traditions is about empirically analyzing teachers' teaching in order to discern which selective traditions are used. This function offers an analytical tool that helps researchers to empirically discern the selective traditions and transform them into a reflection tool for practitioners [8]. With the tool, teachers can individually reflect on their teaching in each educational aspect. Teacher groups can also reflect on their common teaching in extracurricular collaborations and whether they emphasize facts, values or the development of abilities [16]. The tool has also been used to discern the ESE teaching approaches of social science and language teachers [8].

Sund [20] showed in a previous literature review how EE historically developed into ESD in the Swedish context. This earlier review generated five educational aspects (see Figure 1) that show the movement of teachers' educational content from focusing solely on the conveying of facts towards a more pluralistic teaching. The figure shows how the five educational aspects were developed into an analytical tool that included five analytical questions for analyzing teachers' responses in interviews about their ESE teaching. The teachers' responses made three selective traditions visible [21].

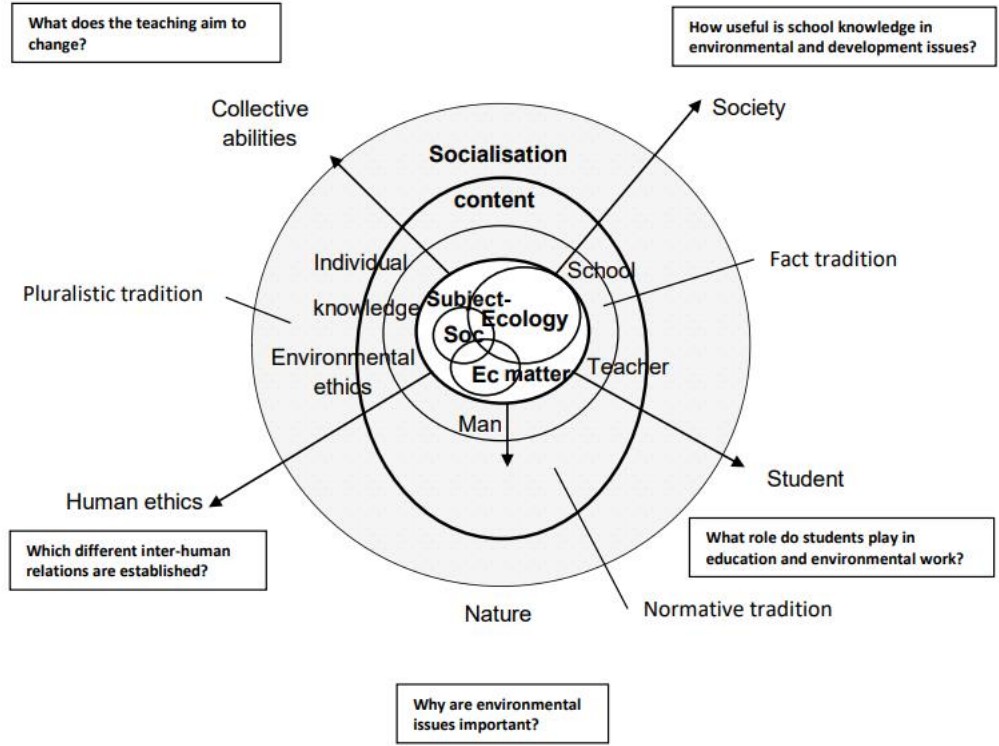

**Figure 1.** The educational content of each selective tradition is 'opened up' through five important *educational aspects*, each of which answers one question. The subject-matter content is consisting of ecological, economical (EC) and social issues (Soc).

Figure 1 shows in a model that the educational content connects more with the surrounding world, outside school, the further it is positioned from the center. This model can be used to analyze ESE teaching empirically. Teachers could themselves, or by others, be positioned in five educational aspects through different accounts of their EE/ESE teaching. The integrated subject matter is placed in the inner circle (shown in bold letters). This circle is the starting point in the left-hand term for each educational aspect, where the fact-based tradition is dominant (next circle outside the subject matter). The normative tradition (bold ellipse) leans more towards nature (biocentrism) and also more outwards in the other aspects. The pluralistic tradition or pluralistic approach (arrows pointing outwards from the center) connects more deeply with the surroundings [21].

### 5.3. Visualizing Longitudinal Changes in ESE Teaching

The concept of selective traditions can function as a way to illustrate or visualize how the emphasis of selective teaching traditions in ESE changes over time. This function is important in that it offers a possibility to visualize the shifts in emphasis in teaching due to changes in the curriculum or other external societal pressures on schools [22].

Table 1 shows the four research studies that analyzed teachers' teaching approaches using the concept of selective traditions. The comparison of results from these four studies makes it possible to recognize that the fact-based tradition became more dominant in the Swedish school context after the curriculum changes in 2011 [23].

**Table 1.** A summary of the distribution of science teachers in three selective traditions in four different studies from 2002 [3], 2011 [21], 2012 [24] and 2020 [8].

| | Fact-Based Environmental Education | Normative Environmental Education | Pluralistic Environmental Education |
|---|---|---|---|
| Swedish National School Agency, 2002 | | | |
| - Lower secondary school (67 teachers) - Questionnaire and interviews - National curriculum 1994 | 11% (7) | 67% (45) | 22% (15) |
| Sund and Wickman, 2011a | | | |
| - Upper secondary school (10 teachers) - Interviews - National curriculum 1994 | 40% (4) | 40% (4) | 20% (2) |
| Borg, Gericke, Höglund and Bergman, 2014 | | | |
| - Upper secondary school (669 teachers) - Large-scale questionnaire - National curriculum 2011 | 40% (268) | 16% (107) | 25% (167) |
| Sund, Gericke and Bladh, 2020 | | | |
| - Lower secondary school (15 teachers) - Questionnaire and interviews - National curriculum 2011 | 54% (8) | 33% (5) | 13% (2) |

The results in Table 1 show an increase in the number of teachers teaching in the fact-based tradition, although several studies are case studies and not generalizable. However, the table shows that the function of selective ESE traditions to analyze and discern the evolvement of ESE teaching changed over time. The trend towards fact-based teaching is also supported by the results of a coming study of Swedish science teachers in lower secondary school [25]. The increase in fact-oriented teaching may be due to the extended core content in the latest national curriculum of 2011 [23]. The teachers involved in the study said that due to the changes in the curriculum they had to focus more on disciplinary concepts and as a result had less time for group discussions or group work [25].

## 5.4. Observing the Distribution of ESE Teaching between School Subjects

The concept of selective traditions can function as a way of observing the distribution of teaching in different subject areas, which can be important extracurricular ESE collaborations. This function offers teacher groups the possibility to discuss the selective traditions that occur in their group and how they are distributed. In order to develop a common teaching approach that offers students a learning environment which embraces facts, values and the development of action competence, there needs to be a variation in the emphasis on different selective traditions. For instance, if all teachers in a collaboration teach in the fact-based tradition, the collaboration could be less fruitful. A variation in selective traditions is thus an important key to success in extracurricular collaborations [16].

In their publication, Borg, Gericke, Höglund and Bergman [6] studied the differences in the distribution of selective traditions among teachers from different subject areas through a large-scale questionnaire study. The emphasis of science teachers' teaching was mostly on the fact-based tradition and that of social science teachers on the pluralistic tradition. Sund, Gericke and Bladh [8] showed that there were some differences in the distribution of the three selective traditions amongst teachers from different subject areas. In this publication, data were gathered from lower secondary school teachers and consisted of responses to a written questionnaire related to analytical questions in order to discern their selective traditions. The science teachers in the study worked in all three selective traditions, whereas the social science teachers mainly worked in the pluralistic tradition. The language teachers in this small sample mostly worked in the normative tradition. Although the sample is small, the results show that science teachers mainly work in the fact-based tradition, while social science teachers work mainly in the pluralistic tradition. Language teachers mainly work in the normative tradition when their teaching is related to sustainability issues [8]. This result is confirmed by those of the previously mentioned large-scale quantitative study [6].

## 5.5. Recognising Tacit Frameworks—Facilitators of and Obstacles to Teaching Outcomes

The concept of selective traditions can function as a way of helping teachers to reflect on their tacit frameworks for teaching. These often unreflected frameworks keep teachers in specific, and often habitual, teaching approaches and can appear as obstacles to change and development. The function of tacit framing is to recognize that teachers' teaching traditions can emphasize teaching that in fact goes against the intention of the curriculum change. The consequences of this can be that pupils do not get the kind of teaching that could make them more successful in national tests. This function points to the fact that teachers need to know what their teaching emphasis is in relation to the curriculum changes on ESE issues, i.e., how they adjust their ESE teaching in an informed and systematic way towards change.

In the reviewed publications, selective traditions can be understood as conceptual schemes of what teachers consider good teaching [21]. Van Driel, Bulte and Verloop [26] used three curriculum emphases to study teachers' domain-specific beliefs about the chemistry curriculum for upper secondary education in the Netherlands. They claim that their study serves as an exemplary case of how teachers' domain-specific beliefs can be investigated and taken into account in the context of educational reform. The study clearly showed that teachers' tacit frameworks can hinder curriculum change. Callahan and Dopico [7] claim that this function is important to recognize in teacher education.

Secondary science teachers' selective traditions were studied by Gyllenpalm, Wickman and Holmgren [27]. The curriculum suggested a more inquiry-oriented approach but, even though the descriptions of the teachers' instructional approaches are varied in the interviews, the knowledge aims are generally similar in that they focus on science subject matter. The selective tradition there was used to describe a teacher's habitual way of conducting inquiries. It is evident that a fact-oriented framework is an obstacle to a more open inquiry. Traditions can also act as barriers to a curriculum supporting ESE teaching when implementing holistic ESD at school [24].

A selective tradition can also be an obstacle to the learning of a science content that is better aligned with the ESE content tested in national tests. Swedish national tests in science include a minimum of 20% socio-scientific issues related to the ESE content. A study of the selective traditions in science teachers' practices and the introduction of national testing show that teachers in the fact-based tradition risk missing important tested content [28]. A selective tradition can also become an obstacle in curriculum change.

*5.6. Showing the Situated and Social Nature of the Existing Selective Traditions*

The concept of selective traditions can function to show the situatedness or context-sensitive nature of teaching. When data are sorted into different categories in empirical research, their variation and complexity are often reduced. This function of the concept shows that teaching is not static and that the teaching context and social peer environment are important.

The complexity of the reality indicates that teachers cannot always be categorized into one selective tradition as it may depend on the teaching context. This was shown in one of the publications, where in the interviews it became apparent that science teachers worked in three different traditions but that they all showed a tendency towards fact-based teaching when describing their practical work [29]. All the teachers focused on teaching scientific facts and skills, and several of them claimed that their conveyance of what was regarded as real knowledge had changed. This result shows that teaching is contextually sensitive and that teaching approaches are not static.

In another publication, a comparison of the results from two studies in which the same teachers participated showed that individual teachers can switch from mainly working within the pluralistic tradition to the fact-based tradition. In the first part of the second study concerning good tasks in national tests [30], science teachers taught the science content according to all three selective traditions [19]. In the second part concerning the teachers' views of what kind of scientific knowledge and abilities students were expected to develop [30], in group discussions, the teachers appeared to work in the fact-based tradition. This result shows the social nature of teaching approaches and that teachers in groups do not emphasize the same selective traditions as they do individually.

*5.7. Promoting Specific Teaching Outcomes*

The concept of selective traditions can function as a theory to promote a specific kind of ESE teaching, most often being the pluralistic teaching tradition. This function highlights the tension between normativity in educational research and practice, and the risk of democratic deficit, which is contradicted between an ESE that tells the student what is right (the normative tradition) and an ESE that aims to provide the student with action competence (the pluralistic tradition). This tension is also related to the needs of the society as contrasted with individuals' emancipation.

In some of the reviewed publications, selective traditions were often used to argue for a specific teaching approach that is suitable for specific reasons. If the long-term purpose of the education is to enhance the development of informed and active young people, conveying factual knowledge is not enough [21]. According to many researchers, ESE could constitute the basis for the development of education for student emancipation and focus on learning in action [10,31]. This means that pupils would need to have educational opportunities to use the knowledge they learn in school in actions outside school [32].

In other publications, the pluralistic tradition embraces democracy [4] and consists of different voices, information, facts and beliefs. In this tradition, values are also important in that they make students aware of the variety of different interests and perspectives. It is important to develop good skills for argumentation in a pluralistic classroom. This is recognized in the international policy debate about ESD that seems to be moving away from a focus on normative behavioral modifications to more democratic pluralistic approaches [33].

## 6. Discussion

This section begins with a discussion about the seven discerned functions of the concept of selective traditions identified in the twenty-four publications in relation to educational philosophy and the ESE research outside the literature included in this systematic review. It continues with discussing the implications of the functions for teacher education and in-service training and gives recommendations for using them in hands-on practice.

### 6.1. The Seven Functions Discussed in Relation to Research Outside This Review Literature

The first function of combining educational philosophy and environmental problems in teaching is useful in discussions about the differences between EE and ESE. The fact-based tradition and the normative tradition are both oriented towards facts and attitudes, whereas the pluralistic tradition is more process-oriented [21,32]. EE teaching is product-oriented in that specific knowledge needs to be learned about how to solve known environmental problems. This can be compared with ESD 1, where the intended learning outcomes are known [14]. The pluralistic tradition makes use of the same educational content, e.g., subject matter, but as a vehicle in the process of developing abilities through discussions and actions for action competence where the solutions for future challenges are open [34,35]. This is comparable with ESD 2, where the solutions for future challenges are still under debate [14].

The second function is analyzing teaching empirically in order to discern which selective traditions are used. The main point about discerning teachers' selective traditions is not to put teachers into different categories. Selective traditions are not static but are situated in the actual teaching context [29,30]. This is important because it indicates that selective traditions can be changed and adjusted. The analysis of teaching approaches contributes to reflective discussions about and possible changes in the teaching. According to Dewey [36], an analysis of teaching does not mean comparing simple behaviors, but rather looking at the more complex approaches developed by teachers' experiences and disciplinary education at university. In this sense, a selective tradition cannot always be explicitly expressed by the teacher but can be discerned through reflection by using an analytical tool (five educational aspects, Figure 1). However, before one can start reflecting on them, it is essential to acknowledge selective traditions as habitual teaching approaches as this will guide one in the search for ways of changing them [37]. The educational aspects [20] can be used by teachers as a reflection tool to discern their own teaching approach [8]. The point is to encourage teachers to start reflecting on their own teaching, preferably together with peers in groups. This type of group reflection by teachers in one subject area or many, in extracurricular collaborations, is a way of developing collaborative ESE teaching [16].

The third function of visualizing longitudinal changes in teaching is important for discerning changes due to a curriculum change or other change pressures on teachers (Sund and Gericke, in review). The identified fact orientation of the teachers' teaching following the latest Swedish curriculum change in 2011 [25] aligns with a Swedish national policy focus on improving the results of PISA surveys which have been decreasing for more than 15 years [30]. The focus of the latest national curriculum is on more easily assessed factual knowledge than open-ended questions or discerning abilities. This is an international phenomenon in the age of measurement [15] and a way of visualizing the entry of neoliberal forces in schools, where almost everything is expected to be measurable [38,39]. This resembles the discussion about EE versus ESD when the United Nations launched the policy process of entering ESD globally [40]. This function also makes researchers and practitioners reflect on what makes their teaching change.

The fourth function of observing the distribution of teaching in collaborations between school subjects is important for developing cross-curricular ESE teaching collaborations. Some teachers are not always happy about this type of collaboration, even though it is promoted in, e.g., the Swedish national curriculum [23]. Most science teachers are rooted in the fact-based selective tradition [8], as Gayford [41] also similarly found. Gayford

further noted that pluralistic thinking seems to be alien to many science teachers as they mostly emphasize the pluralistic tradition [6,42]. This function is threefold in that it can highlight the disciplinary obstacles for collaborations, show the differences in teaching between subject areas in collaborations and indicate how different teaching approaches can complement each other in collaborations. The research has shown that the teaching in different subject areas differs but can together offer students a more comprehensive ESE learning situation [16].

The fifth function of recognizing tacit frameworks—facilitators of and obstacles to teaching outcomes—is important for discerning teachers' conceptual schemes. This has been important in the relation between research and practice. According to Wickman [43], this relationship has historically occurred in three steps: (a) teacher deficit and social engineering, where conceptual schemes are hardly acknowledged, (b) reflecting practitioners, where conceptual schemes aid the choices of already knowledgeable teachers and (c) the mangling of the conceptual schemes by researchers through practice with the purpose of revising research theory. The results of this literature review and study of the concept of selective traditions align with step two, which is close to the didactic model to develop teaching practices and the teaching profession [44]. The authors' experiences are that in discussions with science teachers in in-service training sessions or when teaching student teachers, most teacher groups recognize and are familiar with the concept of selective traditions. Didactic modeling is one way of developing teaching approaches systematically through different models, such as the teaching dimensions of what, how and why by Klafki [45] and curriculum emphases by Roberts [5].

The sixth function showing the situated and social nature of teaching is important for showing how context-sensitive the teaching and selective traditions are. When teachers discussed the importance of practical work in the study by Sund and Wickman [29], they all emphasized the fact-based tradition. This can involve anthropocentric views of nature in excursions and systematically observing the surroundings. Observers are not part of nature, but can be regarded as external observers [46]. Another example of this anthropocentric view is practical work in the laboratory, where nature is manipulated by humans [47]. This function shows that teachers easily embrace certain scientific roles. In one study, when teachers discussed socio-scientific issues in national tests in groups, they all entered into a rational scientific discourse [30]. Östman [48] discovered something similar and explained it as a disciplinary hegemonic discourse that has been common in science teaching since the 17th century. This might look like a historical event, but it can still be a challenge in teacher education. In teacher education, students can often revert to the disciplinary teaching traditions that they learned from others, which can in turn become obstacles in discussions about pluralistic teaching approaches or work in collaborative extracurricular settings [30].

The seventh function of selective traditions is promoting specific teaching outcomes. The promotion can be about developing a more democratic teaching that supports students' development of emancipation and action competence [49–51]. Theoretical discussions inspired by John Dewey [52] concern important aspects of teaching, such as democracy [11,15,53]. The normative tradition is democratically questionable [4]. The democratic participatory approach is a prerequisite for developing pupils' action competences [34]. In teaching practice, research and at the policy level for global development, the learning outcomes of EE/ESD/ESE have increasingly been translated into a number of competences for sustainable development, e.g., critical thinking, collaborative decision-making, future scenario skills and action competence [54]. The underlying educational idea is to empower young people by developing key competencies. Key competencies are something to achieve, whereas action competence is an ongoing teaching approach that encourages pupils to use the knowledge and abilities they have learned at school to guide their actions. Action competence is an educational ideal [34]. Promotion of developed action competence teaching enables pupils to deal with the often-complex societal challenges of sustainable development [35].

*6.2. Implications for Teacher Education and in-Service Training*

The first question to confront Callahan and Dopico [7] when reading about selective traditions was, "Do teachers teach in the same way as they were taught?" If this is the case, we need to analyze the selective traditions that were prevalent when they were studying to become teachers. This is an example of requested further research on selective traditions. Knowing how student teachers are trained in teacher education courses can help us to understand more about how our children will learn about global developmental challenges in the future. The second question for Callahan and Dopico [7] was, "Which part of our teaching is canonical and which is personal input or contributes to the development of universal knowledge?" Learning a discipline is one thing, but teaching it is another. The teachers' disciplinary traditions meet the pupils' everyday knowledge in the classroom. The canonical parts of the discipline meet a transformed school science in the textbooks [55].

It would be fruitful if teacher education institutions could visualize and discuss selective traditions and show how they can work as tacit frameworks for student teachers learning when becoming teachers and also as obstacles to change in school [43,55]. An important question to start asking in teacher education is, "What is new in this curriculum compared to my everyday teaching?" The answer might be a slightly different way of teaching a subject and align towards a selective tradition different from the current practice.

The seven functions of the concept of selective traditions discerned in this review can contribute to a better understanding of how more emancipating, democratic and transforming ESE teaching can be developed. The functions illuminate important qualitative discussions when teaching is developed systematically. Five of the functions are useful in the practice-oriented hands-on development of ESE teaching in teacher education and in-service training, while the other two functions (visualizing and observing) are useful for observing the changes in and distribution of ESE teaching at a school, national and international level.

These functions can be used to develop the teachers' teaching and the learners' learning of skills in alignment with the needs globally. The United Nations Sustainable Development Goals (SDGs) set an agenda for action to contribute to effectively improving life on our shared planet. In effect, they set a policy direction aiming for significant improvements by 2030 [56,57]. Goal 4 attends to the need for quality education for all, and target 4.7 requires that all learners acquire the knowledge and skills needed to promote sustainable development, including explicit education for sustainable development.

## 7. Conclusions

Discussions about how a transformation of teaching occurs (or not) begin with educational philosophies, the root causes of developmental challenges, rational discourses, disciplinary traditions, curriculum changes, external pressures and market forces, all of which are essential for systematic and democratic changes in ESE teaching. Research on the concept of selective traditions has shown that there are many functions to consider when discussing and analyzing ESE teaching for the future in research and practice, as outlined in this review. The seven functions of the selective traditions identified in this study can be a valuable contribution in this endeavor to develop and analyze ESE teaching locally as well as globally in alignment with the SDGs.

**Author Contributions:** The conceptualization, methodology, analysis, and writing of the research article was collectively done by all the authors. All authors have read and agreed to the published version of the manuscript.

**Funding:** This research was supported by the ROSE (Research on Subject-specific Education), Karlstad University.

**Institutional Review Board Statement:** This study is following Swedish research ethical guidelines.

**Informed Consent Statement:** This study is following Swedish research ethical guidelines.

**Data Availability Statement:** See references regarding literature review.

**Conflicts of Interest:** The authors declare no conflict of interest.

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
