# Peer review of "More Than Two Decades of Research on Selective Traditions in Environmental and Sustainability Education—Seven Functions of the Concept"

_sustainability, doi:10.3390/su13126524_

Round 1
Reviewer 1 Report
I applaud the intent of the paper and found the intended topic as a whole intriguing. The abstract does need to make it clear that it is looking purely at the Swedish context, though. Also, there are passages ( eg l.l. 102+) where statements are being made about historical trends with very little citation as proof. Perhaps a diagram or table to clarify periods of change?
My major concern, however, is one of total confusion in overall logic and meaning . On reading the paper, my expectation of how the results would be presented and analysed was not as I interpreted and expected from reading the sections beforehand . It is not clear what the traditions analysed actually are, as named - if they are approaches to teaching ESD, then that did not come through at all. The diagram ( Fig 1) is utterly confusing - how does it relate to the theme being analysed? Why is there no similar diagram for the other trends outline? Are there any other comparative studies elsewhere that could be suggested as a comparison?
Having read the paper several times, and still remaining confused, I believe that problem is a lack of clear description of the method ( how were these themes identified ) and incomplete logical explanation of how Biesta's notion of functions has been translated into... teaching approaches? If the results are meant to identify different teaching approaches, their titles need to be expressed as such, and greater emphasis given to the meaning of the language being used. It is still not evident, for example, by the end of the paper, that is meant by 'selective tradition', and especially, in what sense are they selective?
Author Response
Reviewer 1 (response in italic below)
I applaud the intent of the paper and found the intended topic as a whole intriguing. The abstract does need to make it clear that it is looking purely at the Swedish context, though. Also, there are passages ( eg l.l. 102+) where statements are being made about historical trends with very little citation as proof. Perhaps a diagram or table to clarify periods of change?
Authors’ response: Thank you for your support and interest in our paper. The reviewed studies are not entirely from Sweden, but 22 of 24 are. We have now included a sentence about that in the method section, and also included this fact in the abstract as asked for by the reviewer. We have included references in order to support the writings, and it is important to recognize that this paper do not bring any new empirical data to the table, but is a review referencing and synthesizing previous studies.
My major concern, however, is one of total confusion in overall logic and meaning . On reading the paper, my expectation of how the results would be presented and analysed was not as I interpreted and expected from reading the sections beforehand . It is not clear what the traditions analysed actually are, as named - if they are approaches to teaching ESD, then that did not come through at all. The diagram ( Fig 1) is utterly confusing - how does it relate to the theme being analysed? Why is there no similar diagram for the other trends outline? Are there any other comparative studies elsewhere that could be suggested as a comparison?
Authors’ response: Thank you for noticing this in the text. We agree that there is an inconsistency between the use of the words “approaches” and “teaching traditions”. Therefore, we have deleted the word “approaches” in most instances in the text not to cause any confusion.
Regarding figure 1, the figure stems from one of the included studies in the review study. It is not made in the review-study. We have now included a sentence about that in the figure legend. The figure can be used to analyse the ESE teaching of teachers, as the function states in section 5.2. We describe this shortly in section 5.2, but cannot do that in detail, because there is not room in a review study to give so much specific detail to each of the 24 included articles. Instead, we in our text refer to further reading in the reviewed studies, as recommended for review studies (e.g. see Gough, Oliver, and Thomas, 2017). There was not any such specific figures available in the other 23 studies, that is why no such similar figures are included in the other studies.
Having read the paper several times, and still remaining confused, I believe that problem is a lack of clear description of the method ( how were these themes identified ) and incomplete logical explanation of how Biesta's notion of functions has been translated into... teaching approaches? If the results are meant to identify different teaching approaches, their titles need to be expressed as such, and greater emphasis given to the meaning of the language being used. It is still not evident, for example, by the end of the paper, that is meant by 'selective tradition', and especially, in what sense are they selective?
Authors’ response: Our intention was not to identify new teaching approaches. As responded earlier, in order to avoid this misconception we have deleted most instances relating to “teaching approaches” in the text. Also, we have simplified the abstract, introduction and background, and inserted several sentences explaining that the aim is to identify what functions previous research given the concept of “ESE selective traditions”. Selective traditions as a phenomena is Hopefully, our aim is much better explained now in the text. Also, we have taken the advice from the reviewer and substantially extended the method section describing how the themes were found by dividing the text into two subsections (4.1 and 4.2).
In addition, the reference to Biesta’s functions in the background is shortened. Our study take the departure from the ide of “functions”, but not to educational goals as such (as in the work of Biesta), rather towards functions given to the concept of selective traditions in research. Therefore, we agree that the functions by Biesta might mislead the reader somewhat, and we have substantially shortened that text in the background and discussion referring to Biesta. Also, not to confuse the aim of the study (to identify functions of research on selective traditions) we have deleted all text in the background and discussion referring to “approaches”. The concept of “teaching approaches” are now deleted not to confuse with selective teaching traditions. In this way we hope that our paper is much more clear about what it explores, i.e. to identify functions given to the concept of selective traditions. Thank you for making us aware of this confusion.
Reviewer 2 Report
This is a very clear and well presented manuscript. The topic addressed is important and is one that the authors are well-versed with. The depth of experience researching selective traditions is obvious in the overall clarity of description and discussion in the paper.
The only point where I found potential issue in the paper was the narrow representedness of the 24 reviewed publications. 12 of these publications are ones in which the current authors are also authors, while another 3 have authors who have also published with the current authors. Then there are also 5 or 6 publications by Ohman and Ostman. And finally only 3 publications that seemingly are independent of the core of publications in this group. So if we talked about researchers bringing different views, there are only really about 5 variations in the selected 24 publications.
Author Response
Reviewer 2 (author responses in italic below)
This is a very clear and well presented manuscript. The topic addressed is important and is one that the authors are well-versed with. The depth of experience researching selective traditions is obvious in the overall clarity of description and discussion in the paper.
Authors’ response: Thank you for recognizing the importance of our work!
The only point where I found potential issue in the paper was the narrow representedness of the 24 reviewed publications. 12 of these publications are ones in which the current authors are also authors, while another 3 have authors who have also published with the current authors. Then there are also 5 or 6 publications by Ohman and Ostman. And finally only 3 publications that seemingly are independent of the core of publications in this group. So if we talked about researchers bringing different views, there are only really about 5 variations in the selected 24 publications.
Authors’ response: The reviewer is correct in this observation. We would though like to stress that we used a systematic review approach (see Grant & Booth, 2009), and included all studies that used the framework of ESE selective traditions in their studies. Hence, we have not excluded any studies, but included all studies on the topic. Therefore, no claim can be made that the selection of studies are biased.
Reviewer 3 Report
In this original study, the researchers analyze seven selected concepts of tradition using a qualitative systematic review of a review of previous studies. Maybe that's why the text overlaps by as much as 13%. The study is based on text analysis and
integration of qualitative research and the search for new "topics". Significant links have been identified in as many as twenty-four identified publications. Researchers associate research results with different verbs. Selective traditions that relate to teachers ’attitudes to the content, methods, and goals of education and sustainability education are singled out. It is unfortunate that researchers do not choose the UN Sustainable Development Goals for comparison. Researchers recognize that teachers work largely according to one specific selective tradition, which, as researchers acknowledge, is not recognized by teachers themselves. Discovered by five research, which are valuable for research and practice development. Researchers emphasize that only two of the invented functions are valuable for the study of teaching change. The results are enriched by 20 purposefully selected research, internships, and teacher education practices designed to systematically improve in search of future transformative teaching methods.
It is recommended to print the article, reducing the compliance percentage and linking the findings to the UN Sustainable Development Goals
Author Response
Reviewer 3 (author responses in italic below)
In this original study, the researchers analyze seven selected concepts of tradition using a qualitative systematic review of a review of previous studies. Maybe that's why the text overlaps by as much as 13%.
Authors’ response: We have substantially shortened the manuscript, and thereby reduced the overlaps. Thank you for pointing this out to us.
The study is based on text analysis and integration of qualitative research and the search for new "topics". Significant links have been identified in as many as twenty-four identified publications. Researchers associate research results with different verbs. Selective traditions that relate to teachers ’attitudes to the content, methods, and goals of education and sustainability education are singled out. It is unfortunate that researchers do not choose the UN Sustainable Development Goals for comparison. Researchers recognize that teachers work largely according to one specific selective tradition, which, as researchers acknowledge, is not recognized by teachers themselves. Discovered by five research, which are valuable for research and practice development. Researchers emphasize that only two of the invented functions are valuable for the study of teaching change. The results are enriched by 20 purposefully selected research, internships, and teacher education practices designed to systematically improve in search of future transformative teaching methods.
It is recommended to print the article, reducing the compliance percentage and linking the findings to the UN Sustainable Development Goals
Authors’ response: As pointed out in our previous comment (see above) we have shortened the manuscript, both in background and discussion, to reduce the redundancy of the text. Also, at the end in the discussion we have included a section relating the results to the UN Sustainable Development Goals.
Round 2
Reviewer 1 Report
I congratulate the authors on the ways that they have responded to the initial reviews. This is a much improved and valuable contribution to the literature.